# Jatrolignans C and D: New Neolignan Epimers from *Jatropha curcas*

**DOI:** 10.3390/molecules27113540

**Published:** 2022-05-31

**Authors:** Yi-Lin He, Pei-Zhi Huang, Hong-Ying Yang, Wei-Jiao Feng, Zhao-Cai Li, Kun Gao

**Affiliations:** 1State Key Laboratory of Applied Organic Chemistry, College of Chemistry and Chemical Engineering, Lanzhou University, Lanzhou 730000, China; heyl18@lzu.edu.cn (Y.-L.H.); huangpzh21@lzu.edu.cn (P.-Z.H.); yanghy19@lzu.edu.cn (H.-Y.Y.); fengwj20@lzu.edu.cn (W.-J.F.); 2Research Institute, School of Biological and Pharmaceutical Engineering, Lanzhou Jiaotong University, Lanzhou 730070, China; 3State Key Laboratory of Veterinary Etiological Biology, Lanzhou Veterinary Research Institute, Chinese Academy of Agricultural Sciences, Lanzhou 730046, China; lizhaocai@caas.cn

**Keywords:** Euphorbiaceae, *Jatropha c**urcas* L., neolignan, jatrolignans, epimers, antichlamydial activity

## Abstract

Two new neolignans jatrolignans, C (**1**) and D (**2**), a pair of epimers, were isolated from the whole plants of *Jatropha c**urcas* L. (Euphorbiaceae). Their structures were determined with HRESIMS, IR, and NMR data analysis, and electronic circular dichroism (ECD) experiments via a comparison of the experimental and the calculated ECD spectra. Their antichlamydial activity was evaluated in *Chlamydia abortus.* They both showed dose-dependent antichlamydial effects. Significant growth inhibitory effects were observed at a minimum concentration of 40 μM.

## 1. Introduction

*Jatropha c**urcas* L. has been used as a traditional medicine for the treatment of traumatic injury, fracture, itchy skin, eczema, and acute gastroenteritis [1]. Its extracts and monomeric compounds possess potential pharmacological activities, owing to the efficiency of clearing heat and detoxication scattered stasis detumescence [1]. Chemical constituent investigations of the roots, stems, and leaves of *J. c**urcas* led to the identification of structurally diverse diterpenoids [2], triterpenes [3], lignans [4], and coumarins [5], and many of these compounds exhibited promising cytotoxicity [6], antitumor [7], antimicrobial [8], cytopathic [9], anti-inflammatory [10], antioxidant [11], anticoagulant [12], insecticidal [13], and molluscicidal [14] activities. As a part of our ongoing research program for the discovery of potential pharmacological ingredients from natural products, we studied a methanol extract from the dried whole plant of *J. c**urcas;* and in the process, two undescribed neolignan epimers (compounds **1** and **2**) were isolated. Structurally, **1** and **2** (Figure 1) possessed the same planar structure, from which could be speculated a pair of epimers at either C-7 or C-8. The relative configurations of **1** and **2** were determined by comparing their coupling constants between H-7 and H-8, and the absolute configurations were deduced from the ECD spectra of **1** and **2**. We, herein, report the details of the isolation and structural elucidation of **1** and **2,** as well as their antichlamydial activity.

## 2. Results and Discussion

### 2.1. Structure Elucidation

Compound **1** was obtained as an amorphous powder, and it possessed a molecular formula of C_25_H_30_O_9_ on the basis of ^13^C NMR and HRESIMS data (*m*/*z* 497.1783, [M + Na]^+^) (calcd for C_25_H_30_O_9_Na, *m*/*z* 497.1782), which indicated 9 degrees of hydrogen deficiency. Its IR spectrum showed absorption bands consistent with the presence of hydroxy (3452 cm^−1^), carbonyl (1736 cm^−1^), alkenyl (1457 cm^−1^), and aromatic ring (799 cm^−1^) functionalities [15]. The ^1^H NMR spectrum of **1** (Table 1) showed the typical spin systems of two sets of 1,2,4-substituted aromatic ring, in which the characteristic signals were observed at 6.91 (d, *J* = 1.6 Hz, H-2), 6.87 (d, *J* = 8.2 Hz, H-5), and 6.86 (dd, *J* = 8.2, 1.6 Hz, H-6); 6.85 (d, *J* = 1.6 Hz, H-2′), *δ*_H_ 6.67 (d, *J* = 8.3 Hz, H-5′), and 6.81 (dd, *J* = 8.3, 1.6 Hz, H-6′). In addition, a pair of *trans* double bond protons at *δ*_H_ 6.54 (d, *J* = 15.9 Hz, H-7′) and 6.14 (dt, *J* = 15.9, 6.6 Hz, H-8′), and three methoxy groups at *δ*_H_ 3.78 (s, 3-OCH_3_), 3.28 (s, 7-OCH_3_), and 3.84 (s, 3′-OCH_3_) were also observed. The ^13^C NMR (Table 1) and HSQC data of compound **1** exhibited 25 carbon signals, which were classified as two carbonyl carbons, six quaternary carbons (six sp^2^ hybridized carbons), ten methines (eight sp^2^ hybridized carbons and two oxygenated carbons), two methylenes, and five methyls. They were assigned as two 1,2,4-substituted benzene rings, two double bond carbons, three methoxyl groups (*δ_C_* 57.3, 56.1, 55.9), two oxymethylene carbons (*δ_C_* 65.3, 63.8), two oxymethine carbons (*δ_C_* 82.6, 82.1), and two acetyl groups (*δ_C_* 21.2, 171.0 and *δ_C_* 21.0, 171.0).

The ^1^H-^1^H COSY correlations of **1** between H-7/H-8/H_2_-9 and H-5/H-6, as well as the HMBC (Figure 2) correlations from H-7 to C-1/C-2/C-6/C-8/C-9/OCH_3_, H-2 to C-1/C-3/C-6/C-7, H-5 to C-1/C-3/C-6, H-6 to C-1/C-3/C-5, H-8 to C-7/C-9 and H_2_-9 to C-7/C-8/OA_C_, indicated the presence of a guaiacyl glycerol moiety in C-1 [16]; the ^1^H−^1^H COSY correlations between H-7′/H-8′/H_2_-9′ and H-5′/H-6′, as well as the HMBC correlations from H-2′ to C-1′/C-3′/C-4′/C-6′/C-7′, H-5′ to C-1′/C-3′/C-4′/C-6′, H-6′ to C-2′/C-4′/C-7′, H-7′ to C-1′/C-2′/C-6′/C-9′, H-8′ to C-1′/C-9′ and H_2_-9′ to C-7′/C-8′/OAc, indicated the presence of a coniferyl alcohol moiety in C-1′. Two phenylpropanoid units in **1** were connected through an ether bond by the HMBC correlation from H-8 to C-4′, which indicated **1** as being a neolignan structural type of 8-4′ [17]. The positions of the two methoxy groups were determined at C-3 and C-3′, respectively, due to the HMBC correlations of 3-OCH_3_ (*δ* 3.78)/C-3 (*δ* 150.9) and 3′-OCH_3_ (*δ* 3.84)/C-3′ (*δ*146.7). Thus, the substitution patterns of two 1,2,4-substituted aromatic rings were confirmed.

The vicinal coupling constant of H-7/H-8 can be used to assign *erythro* versus *threo* relative configurations [18,19]. The open-china *erythro* isomer generally had smaller coupling constants than the open-china *threo* isomer in non-hydrogen bonding solvent. The coupling constant (8.8 Hz) of H-7/H-8 indicated a *threo* sterostructure for compound **1**.

The absolute configuration of **1** was proposed as depicted, based on the calculated ECD curve, which agreed well with the experimental ECD data (Figure 3), allowing the absolute configuration of **1** to be defined as 7*S* and 8*S*. Hence, the structure of **1** was designated and named jatrolignan C [20].

Compound **2** was also obtained as an amorphous powder. The molecular formula was established as C_25_H_30_O_9_ (9 degrees of unsaturation) from its HRESIMS (*m*/*z* 497.1785, [M + Na]^+^) (calcd for C_25_H_30_O_9_Na, *m*/*z* 497.1782). Its IR spectrum showed absorption bands consistent with the presence of hydroxy (3436 cm^−1^), carbonyl (1738 cm^−1^), alkenyl (1453 cm^−1^), and aromatic ring (767 cm^−1^) functionalities. The ^1^H and ^13^C NMR signals (Table 1) of **2** were almost identical to those of **1**. The discriminating coupling constants of H-7, H-8 and H_2_-9 indicated the relative stereochemistry of **2** was different from **1**, thus **2** was suggested to be the epimer of **1** at C-7 or C-8 and *erythro* sterostructure, which was confirmed by the ECD spectra (Figure 3), indicating that **2** gave an exactly opposite Cotton effect at 220 nm compared with that of **1**. The absolute configuration of **2** was proposed as depicted, based on the calculated ECD curve, where the calculated values of 7*R* and 8*S* matched the experimental ECD curve (Figure 3), allowing the absolute configuration of **2** to be defined as 7*R*, 8*S*. Hence, the structure of **2** was designated and named jatrolignan D.

To determine whether compounds **1** and **2** were natural or artificial products, the MeOH extract of *J**. c**urcas* was subjected to an MCI gel column with MeOH/H_2_O (80%), applied to Sephadex LH-20 (MeOH), and then compared to the isolated compounds **1** and **2** using HPLC. The HPLC (Appendix A) showed that the preliminary extract contained compounds **1** and **2**, indicating that the both **1** and **2** are a metabolites of the plant.

### 2.2. The Antichlamydial Activity of Compounds

Chlamydial infections in humans and animals are global health issues. Although chlamydial infections within the human population are currently manageable with the existing conventional therapies (antibiotics treatment), the extended exposure of Chlamydia to antibiotics provides greater opportunity for the development of antibiotic resistance in chlamydial species. Natural products show significant potential for treating chlamydial infections, which is expected to produce new antichlamydial treatment modalities. Neolignans have shown multiple activities, such as anticarcinoma, antioxidation, and anti-HIV effects. In order to find the new medicinal potential of neolignans, the antichlamydial activity of two novel neolignans, compounds **1** and **2,** from the medicinal herb *Jatropha c**urcas* L. was evaluated in this study, which might reveal a new potential antichlamyidal agent for drug development. 

*Chlamydia* spp. are a group of obligated intracellular bacteria associated with major diseases in humans and animals. In this study, the antibacterial activities were investigated in Chlamydia abortus, an important zoonotic chlamydial pathogen. Compounds **1** and **2** showed a similar antichlamydial effect on Chlamydia abortus, in a dose-dependent manner. As shown in Figure 4A, with the increasing concentration of compound **1**, the intracellular chlamydial inclusions were smaller in size and less in number. At the highest concentration of 80 μM, inclusions were few and tiny, analogous to the positive control tetracycline (final concentration, 5 μM). A similar effect of compound **2** on chlamydial inclusions of Chlamydia abortus was also observed. The inclusion formation ratio was significantly reduced in cell cultures treated with compounds **1** and **2** at a concentration of 40 μM or more (Figure 4B,C).

## 3. Experimental Section

### 3.1. General Experimental Procedures

Optical rotation was performed on an A RUDOLPH AUTOPOL IV polarimeter (Rudolph Research Analytical, Madison, WI, USA). The UV spectra were recorded on a Shimadzu UV-260 spectrophotometer (Shimadzu Corporation, Tokyo, Japan). The IR spectra were obtained from a Bruker TENSOR27 spectrometer (Rudolph Research Analytical, Karlsruhe, Baden-Württemberg, Germany). The HRESIMS data were obtained on a Thermo Scientific LTQ-Orbitrap Elite-ETD MS spectrometer (Thermo Fisher Scientific, Waltham, MA, USA). Electronic circular dichroism (ECD, JASCO Corporation, Hachioji-shi, Tokyo, Japan) curves were recorded with an Olis DSM-1000 spectrometer using MeOH as solvent. ^1^H, ^13^C, and 2D NMR spectra were run on a Bruker AVANCE III-500/NEO-600 spectrometer (Rudolph Research Analytical, Madison, WI, USA), at room temperature. The ^1^H chemical shifts (*δ*_H_) and ^13^C chemical shifts (*δ*_C_) were measured in ppm, relative to CDCl_3_. Semipreparative HPLC was performed on a Shimadzu LC-10AVP liquid chromatograph, with a YMC-pack C18 (ODS) column (10 × 250 mm, 10 μm, Tokyo, Japan). Column chromatography (CC) was performed on Silica gel (200–300 mesh; Qingdao Marine Chemical Co., Qingdao, China), GE Sephadex LH-20 (GE Healthcare Bio-Sciences, Uppsala, Sweden), and MCI gel CHP 20P (75–150 μm, Mitsubishi Chemical Corp., Tokyo, Japan) and ODS (50 μm, YMC). Silica gel GF254 plates (Qingdao Haiyang Chemical Group Corp., Qingdao, China) were used for TLC.

### 3.2. Plant Materials

Whole plants of *Jatropha curcas* L. were collected in October 2018 from Hainan Province, China, and identified by Associate Researcher Dao-Geng Yu of the Chinese Academy of Tropical Agricultural Science, with a voucher specimen (No. JA20181012) being deposited in the State Key Laboratory of Applied Organic Chemistry, Lanzhou University.

### 3.3. Extraction and Isolation

Air-dried whole plants of *Jatropha curcas* L. (3.0 kg) were extracted with MeOH (3 × 50 L) at room temperature. The solvent was evaporated to produce a residue (99 g) that was suspended in H_2_O and sequentially partitioned with petroleum ether, EtOAc, and *n*-BuOH to yield petroleum ether-, EtOAc-, *n*-BuOH-, and H_2_O-soluble fractions, respectively. The EtOAc- and *n*-BuOH-soluble fractions were separated on a macroporous resin column (MeOH/H_2_O, 0:100, 30:70, 50:50, 80:20, and 100:0, *v*/*v*) to yield five fractions (Fr. A−Fr. E), respectively. Fr. D (20 g) were subjected to MCI column chromatography and eluted with a gradient system of MeOH/H_2_O (from 0:100 to 100:0) to yield ten subfractions (Fr. D1–10). Fr. D5–8 were separated by column chromatography over silica gel (CH_2_Cl_2_/MeOH, from 100:0 to 0:100) to yield 20 fractions (Fr. D. A1–20). Fr. D. A3–5 (3.7 g) was applied to Sephadex LH-20 (MeOH) columns to yield 15 fractions (Fr. D. A. B1–15). Fr. D. A. B8 (96 mg) was chromatographically separated using reversed-phase semipreparative HPLC (C_2_H_3_N/H_2_O, 6/4, *v*/*v*, flow rate, 2.0 mL/min) to afford compounds **1** (3.2 mg) (t_R_ = 26 min) and **2** (2.8 mg) (t_R_ = 24 min).

#### 3.3.1. Jatrolignan C (**1**)

An amorphous powder; [*α*${]}_{D}^{25.6}$ −2.0(C 0.5.CH_2_Cl_2_); UV (MeOH) λmax (log ε): 266.0, 220.0 and 214.0 nm; IR (KBr) *ν_max_* 2961, 1736, 1603, 1511, 1457, 1370, 1260, 1092, 1028, 965, and 799 cm^−1^; HRESIMS (*m*/*z* 497.1783, [M + Na]^+^) (calcd for C_25_H_30_O_9_Na, *m*/*z* 497.1782); ^1^H NMR (600 MHz, CDCl_3_) and ^13^C NMR data (125 MHz, CDCl_3_), see Table 1.

#### 3.3.2. Jatrolignan D (**2**)

An amorphous powder; [*α*${]}_{D}^{25.6}$ −2.91 (c 0.8, CHCl_3_); UV (MeOH) λmax (log ε): 266 220.0 and 214 nm; IR (KBr) *ν_max_* 2937, 1737, 1601, 1511, 1453, 1368, 1236, 1098, 1033, 964, and 787 cm^−1^; HRESIMS (*m*/*z* 497.1785, [M + Na]^+^) (calcd for C_25_H_30_O_9_Na, *m*/*z* 497.1782); ^1^H NMR (600 MHz, CDCl_3_) and ^13^C NMR data (125 MHz, CDCl_3_), see Table 1.

### 3.4. ECD Calculation

The ECD calculations of **1** and **2** were carried out using previous methods. A detailed description of this section is provided in the Appendix A.

### 3.5. Chlamydia Strains and Cell Line

The zoonotic intracellular bacterium *Chlamydia abortus* strain GN6 used in this study was cultured in the mouse embryonic fibroblast cell line McCoy, as described previously (PMID: 33065117).

### 3.6. Antichlamydial Activity Screening

To test the antichlamyidal activity, a concentration of 0 μM (0.5% DMSO as vehicle) to 80 μM of each compound was added in the medium. Tetracycline of 5 μM final concentration was used as a positive control. The *Chlamyida inocula* were incubated with 1 × 106 McCoy cells per well in a 6-well plate. After centrifugation, inocula were replaced with chlmaydial growth medium (RPMI-1640 medium supplemented with 5% fetal bovine serum (FBS), 100 U/mL of kanamycine, 100 μg/mL of streptomycin, and 1 μg/mL of cycloheximide) with 0 μM to 80 μM of the tested compound added, and then incubated in a 5% CO_2_ incubator at 37 °C for 48 h. Afterwards, the chlamydial inclusions were visualized by immunofluorescence staining, using a *Chlamydia abortus* specific mouse anti-MOMP monoclonal antibody as the primary antibody. The inclusion formation ratio (expressed as the number of inclusions/number of cells × 100%) was calculated in the cell cultures [21].

## 4. Conclusions

Jatrolignans C and D, two new neolignan epimers were isolated from the whole plants of *Euphorbiaceae Jatropha curcas* L. The absolute configurations of Jatrolignans C and D were accurately elucidated by means of spectroscopic techniques, especially an extensive NMR data analysis and ECD calculation. They exhibited weak antichlamydial activity compared to tetracycline, which was used as a positive control. To the best of our knowledge, this is the first report to evaluate the antichlamydial activity of neolignans. As components of *Jatropha curcas* L, more detailed chemical and biological investigations of the plant metabolites are required to determine their contribution to supporting and enhancing the application of herbal medicines.

## Figures and Tables

**Figure 1 molecules-27-03540-f001:**
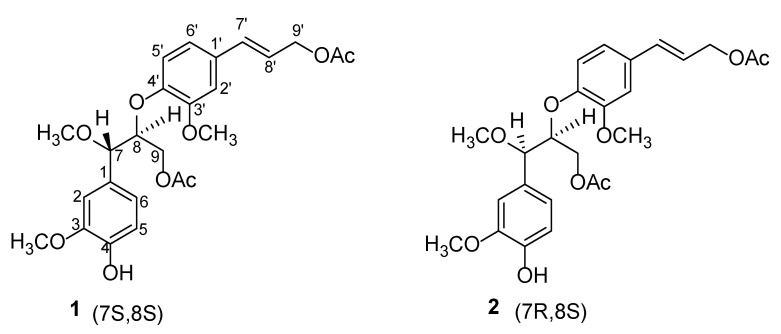
Structures of compounds **1****, 2**.

**Figure 2 molecules-27-03540-f002:**
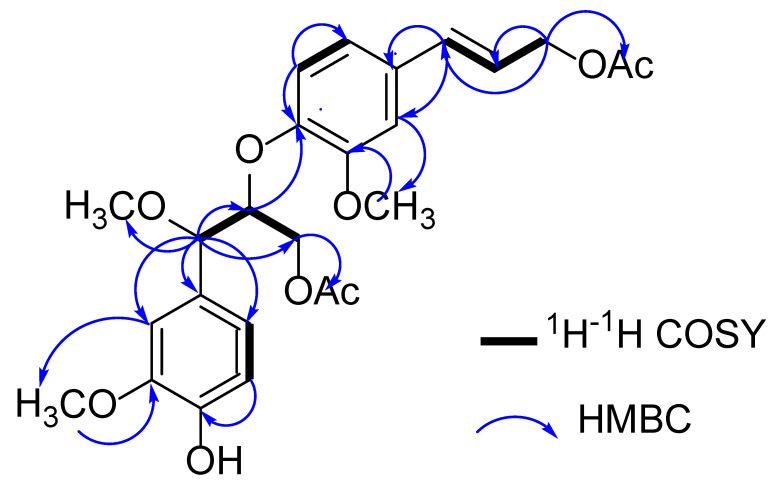
^1^H^−1^H COSY (black bold), key HMBC (blue arrows) correlations of compounds **1–2**.

**Figure 3 molecules-27-03540-f003:**
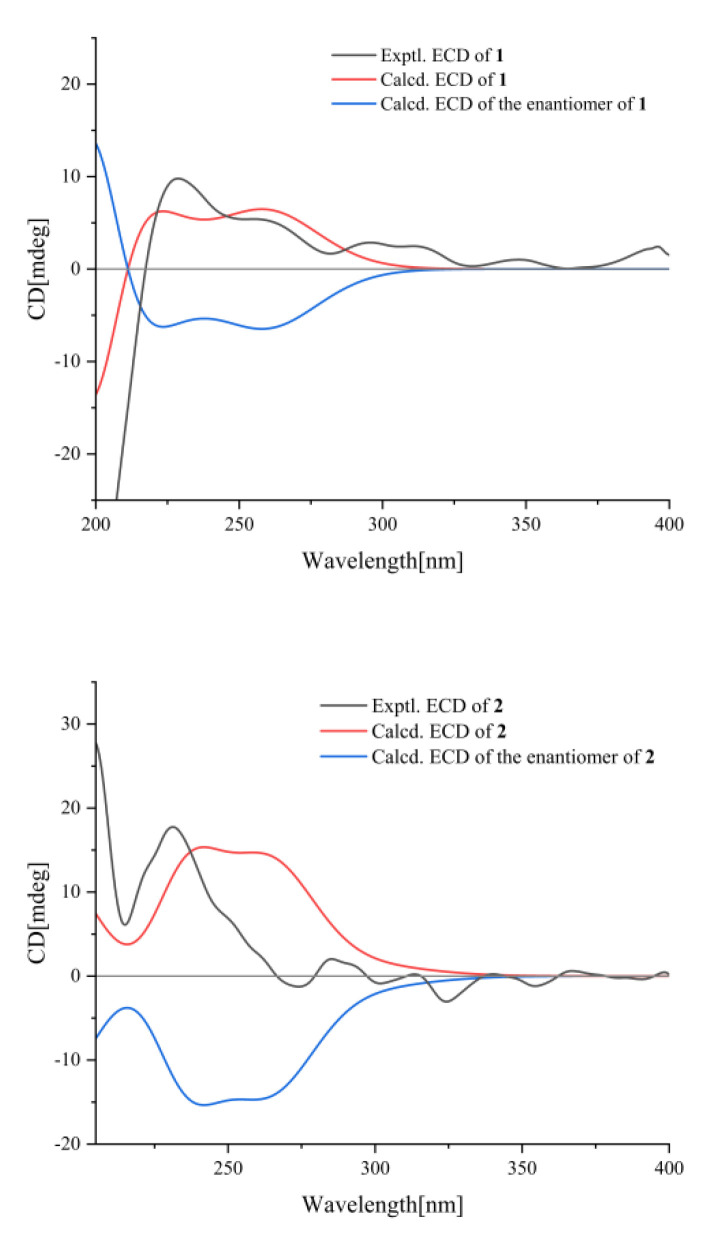
The ECD spectra of compounds **1****, 2**.

**Figure 4 molecules-27-03540-f004:**
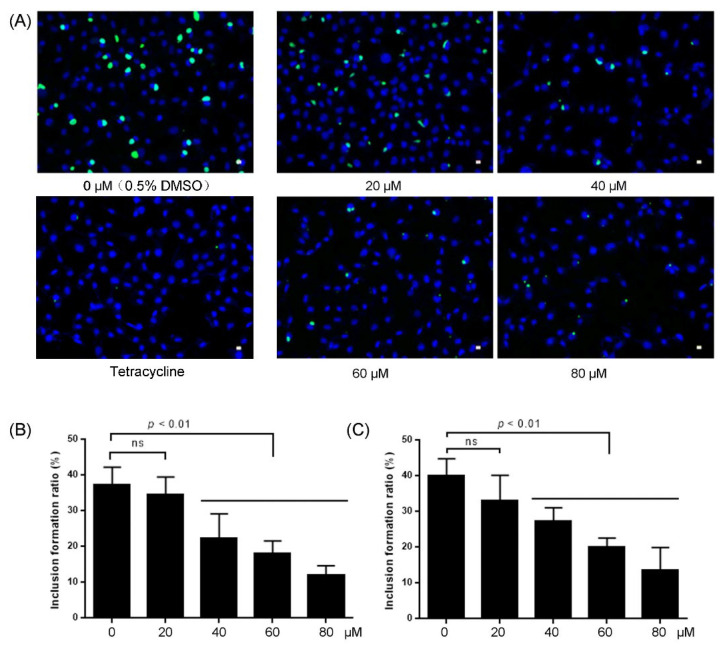
Dose-dependent antichlamydial effects of compounds **1** and **2**. *Chlamydia abortus* strain GN6 cultured in McCoy cells were treated with various concentrations of compound **1** or **2**. Tetracycline (final concentration of 5 μM) was used as a positive control. The chlamydial inclusions were visualized by immunofluorescent staining, and the inclusion formation ratio was utilized to represent the antichlamydial activities. (**A**) *Chlamydia abortus* inclusions were smaller in size and less in number in cell cultures treated with compound **1**. (**B**) A significant reduction of the inclusion formation ratio of *Chlamydia abortus* in cell cultures treated with compound **1**. (**C**) A significant reduction of the inclusion formation ratio of *Chlamydia abortus* in cell cultures treated with compound **2**. ns, no significant difference; *p* < 0.01, significant difference.

**Table 1 molecules-27-03540-t001:** The ^1^H (600 MHz) and ^13^C NMR (150 MHz) data for **1**, **2** in CDCl_3_.

	1		2		
Position	*δ* * _C_ *	*δ*_H_ (*J* Hz)	*δ* _C_	*δ*_H_ (*J* Hz)	HMBC
1	130.0 s		129.5 s		
2	110.0 d	6.91 d (1.6)	109.6 d	6.93 d (1.5)	H-2/C-1,3,6,7
3	150.9 s		150.7 s		
4	145.6 s		145.7 s		
5	114.1 d	6.87 d (8.2)	114.1 d	6.88 d (8.5)	H-5/C-1,3,6
6	121.1 d	6.86 dd (8.2, 1.6)	120.8 d	6.84 dd (8.5, 1.5)	H-6/C-1,3,5
7	82.6 d	4.39 d (8.8)	83.2 d	4.42 d (5.8)	H-7/C-1,2,6,8,9/OCH_3_
8	82.0 d	4.46 m	81.6 d	4.50 ddd (6.1,5.8,4.0)	H-8/C-7,9,4′
9	63.8 t	4.42 m	63.8 t	4.05 dd (11.8, 6.1)	H_2_-9/C-7,8,OA_C_
		4.42 m		4.22 dd (11.8, 4.0)	
1′	131.2 s		130.9 s		
2′	110.2 d	6.85 d (1.6)	109.9 d	6.86 d (1.5)	H-2′/C-1′,3′,4′,5′,6′,7′
3′	146.7 s		146.8 s		
4′	148.1 s		148.6 s		
5′	118.4 d	6.67 d (8.3)	117.8 d	6.91 d (8.5)	H-5′/C-1′,3′,4′,6′
6′	119.9 d	6.81 dd (8.3, 1.6)	119.8 d	6.88 dd (8.5, 1.5)	H-6′/C-2′,4′,7′
7′	134.2 d	6.54 d (15.9)	134.2 d	6.58 d (15.7)	H-7′/C-1′,2′,6′, 9′
8′	121.9 d	6.14 dt (15.9, 6.6)	121.7 d	6.16 dt (15.7, 6.6)	H-8′/C-1′, 9′
9′	65.3 t	4.69 dd (6.6, 1.3)	65.2 t	4.71 d (6.6)	H_2_-9′/C-7′,8′,OAc
		4.69 dd (6.6, 1.3)		4.71 d (6.6)	
7-OCH_3_	57.3 q	3.28 (s)	57.1 q	3.28 (s)	CH_3_/C-7
3-OCH_3_	55.9 q	3.78 (s)	55.8 q	3.84 (s)	CH_3_/C-3
3′-OCH_3_	56.1 q	3.84 (s)	56.0 q	3.87 (s)	CH_3_/C-3′
9-OAc	21.0 q, 171.0 s	2.01 (s)	20.8 q, 170.7 s	1.97 (s)	CH_3_/C=O
9′-OAc	21.2 q, 171.0 s	2.09 (s)	21.0 q, 170.9 s	2.10 (s)	CH_3_/C=O

## Data Availability

Not applicable.

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
