# Peer review of "Jatrolignans C and D: New Neolignan Epimers from Jatropha curcas"

_molecules, 2022, doi:10.3390/molecules27113540_

Round 1

Reviewer 1 Report

The MS titled ``A pair of lignan epimers from Jatropha carcass`` reports the isolation of two new neolignans along with their antichlamydial activity

The MS can not be considered for publication in its current form, the below issues should be addressed

Title

I suggest the title should be changed to ``Jatrolignans C and D: New neolignan epimers from Jatropha carcass``.

Abstract

-The plant family name should be added.

- Chlamydia abortus should be italic through the whole MS.

- Type of assay should be added.

-Positive control and its results should be added.

Keywords

-Add plant family name

-Add Jatrolignans

Introduction

-It is very weak. More details about the traditional medicinal uses of the plant should be added.

- ``Extracts possess potential pharmacological activities``, this should be highlighted in the MS.

  1. Results and discussion

- It is necessary to cover this point. Since the authors used ethyl acetate in the extraction process, therefore the presence of the acetyl group could be from solvent and these compounds could be artifact products. To remove this doubt author should carry out HPLC for the MeOH extract along with the two compounds for confirming their original presence in the extract. The HPLC charts should be added in the supplementary and the result should be discussed in the MS.  

-`` which indicated 6 degrees of hydrogen deficiency``, this should be 9 degrees of hydrogen deficiency???, check.

-``Its IR spectrum showed absorption bands….``, a reference should be added.

-``indicated the presence of a guaiacyl glycerol moiety`` a reference should be added.

-The H-H COSY and HMBC correlation that confirmed the presence of 1,2,4-substituted benzene rings and the position of methoxy groups should be discussed.

- The Key HMBC correlations that confirm the attachment of guaiacyl glycerol moiety to the 1,2,4-substituted benzene ring should be discussed.

- ``The substitution patterns of two aromatic rings were confirmed by the following NOESY ……….``, the substitution pattern should be confirmed by HMBC not NOESY.

_Where are the NOESY correlations of the guaiacyl glycerol moiety, because it is the stereo-containing part, not the benzene ring>

- Please, check the way of citing tables and figures in the MS from any previously published article in Molecules journal.

Table 1

-The multiplicity of all carbons should be added.

-HMBC correlations should be added.

- More details in the discussion of compound 2 should be added.

2.2. The antichlamydial activity of compounds

- ``Chlamydia,`` Chlamydia abortus`` .

-Paragraph about antichlamydial natural metabolites and their significance should be included in the MS.

-why were the authors selected to evaluate antichlamydial for the isolated metabolites, is this activity previously reported for the plant or any of its reported metabolites?. This should be discussed in the MS.

-Urgently, a positive control (reference antichlamydial agent`` should be included in this experiment along with its results to confirm the effect of tested compounds, otherwise judging the effectiveness of these metabolites will be insignificant.

  1. The Experimental Section

-should be corrected to Experimental Section.

-For the plant material voucher specimen number should be added.

- A reference for the antichlamydial activity screening should be added.

Conclusions

It is similar to the abstract and should be modified, the significance of this work, recommendation and perspective should be included.

Reviewer 2 Report

This manuscript by Yi-Lin He et al. deals with a characterization of two diastereomers, namely jatrolignans, isolated from plant Jatropha Curcas.

Absolute configuration of these two new lignans was assigned by means of NMR (which allowed to assign relative configurations) and ECD chiroptical spectroscopy combined to quantum-mechanics calculations.

I find this manuscript with clear scientific motivation and suitable for publication after some revisions, clarifications and improvement of the AC assignment.

1) In paragraph 2.1 the authors state that IR band at 2961 cm-1 is allied to hydroxy group: the 2900 ca. bands are related to C-H stretchings; actually the OH stretching mode is located at ca. 3500 cm-1 as depicted in Figures SI8 and SI16. Please revise.

2) The authors should add the experimental and calculated comparison also for the UV absorptions for compounds 1 and 2. The two ECDs (for 1 and 2) are quite similar but the CD band at ca. 200 nm: were the solutions in the right maximum allowed absortpion range? UV and ECD should be also reported in epsilon and delta-epsilon units respectively. Also there are no information about used concentrations, pathlength cuvette and number of ECD-UV scans accumulation.

3) Computational investigation would be much more complete if the authors added Boltzmann's distribution details about the averaged conformers in ECD calculations (number of conformers, energies difference with respect to the most stable, population percentages). These molecules possess several degrees of freedom and it would be interesting to see how conformers are populated according to their relative energies.

5) Concerning the optimized conformers involved in ECD prediction: are they real minima? Did the authors check for any immaginary frequencies in calculated geometries? According to computational methods description in Supporting Material, the authors performed two different level of theory for optimizations and for frequencies calculation: this could drive to not real minima geometries.

6) According to computationals description in Supporting Material, the authors performed geometries optimizations in vacuo, then they performed ECD calculations with a solvation model; methanol is a polar solvent and the relative stability of the involved conformers may change if their relative energies are calculated in vacuo or with a solvation model. This could lead to different Boltzmann's distributions and to different calculated averaged ECD spectra (depending, of course, on the difference between single conformers ECD spectra to be weighed). Did the authors check Boltzmann's distribution also in solvent model approach?

Some minor considerations:

  • according to all cited references I got that the proper plant name is Jatropha curcas: the authors should revise 'carcas' in manuscript title, keywords, abstract and in the main text;
  • in page 2, line 69, "non-hydrogen solvents" should be revised as "non-hydrogen bonding solvent" or "protic solvents";
  • in page 4, lines 86-87, the authors refer to a CD band (or Cotton effect) as an "absorption band". Please revise.

Round 2

Reviewer 1 Report

No comments

Reviewer 2 Report

The manuscript is suitable for publication after some minor revision.

- Please revise with 'curcas' in:

page 4 line 122 (carcass); page 1 line 22 (curcass).

- In page 2 line 76 please revise with non-hydrogen bonding solvent (as reported in cover letter).

- In page 4 line 99 please revise with Cotton effect (with capital C), or CD band, instead of cotton effect.